# LZER0: A Cost-Effective Multi-Purpose GNSS Platform

**DOI:** 10.3390/s22218314

**Published:** 2022-10-29

**Authors:** David Zuliani, Lavinia Tunini, Marco Severin, Michele Bertoni, Cristian Ponton, Stefano Parolai

**Affiliations:** 1National Institute of Oceanography and Applied Geophysics—OGS, 33100 Udine, Italy; 2SoluTOP SAS, 33050 Pavia di Udine, Italy

**Keywords:** GNSS, cost-effective, u-blox, monitoring, real-time, RTKLIB, Raspberry Pi, landslide, cadastral, automotive

## Abstract

Recent advances in Global Navigation Satellite System (GNSS) technology have made low-cost sensors available to the mass market, opening up new opportunities for real-time ground deformation and structure monitoring. In this paper, we present a new product developed in this framework by the National Institute of Oceanography and Applied Geophysics–OGS in collaboration with a private company (SoluTOP SAS): a cost-effective, multi-purpose GNSS platform called LZER0, suitable not only for surveying measurements, but also for monitoring tasks. The LZER0 platform is a complete system that includes the GNSS equipment (M8T single-frequency model produced by u-blox) and the web portal where the results are displayed. The GNSS data are processed using the RTKLIB software package, and the processed results are made available to the end user. The relative positioning mode was adopted both with real-time and post-processing RTKLIB engines. We present three applications of LZER0—cadastral, monitoring, and automotive—which demonstrate that it is a flexible, multi-purpose platform that is easy to use in terms of both hardware and software, and can be easily deployed to perform various tasks in the research, educational, or professional sectors.

## 1. Introduction

Global Navigation Satellite Systems (GNSSs) provide a worldwide dataset that is vital for a variety of applications, such as cadastral surveying, automotive engineering, crustal deformation monitoring, geodetic reference frame analysis, and even meteorology. Modern GNSS receivers are capable of tracking different satellite constellations (GPS, GLONASS, Galileo, BeiDou, and SBAS) and using different carrier frequencies (e.g., for GPS, L1, L2, and L5), allowing accuracies in the order of millimeters to be achieved, especially when dual-frequency devices are used that can track at least two carrier frequencies (e.g., L1 and L2 GPS signals). However, the high cost of this type of equipment limits its application in monitoring projects when a large number of measurement points are required. Fortunately, in recent decades, significant advances in GNSS technology have enabled the development of low-cost, single-frequency sensors (i.e., u-blox EVK-5T, LEA-4T, LEA-6T, NEO-7P, M8T, and EVK-6T) that are attracting the scientific community’s interest, and several studies have been conducted showing that equipment built with them is capable of centimeter-scale positioning [1,2,3,4,5,6]. A recent study also reported the excellent performance of the cost-effective, dual-frequency u-blox F9P receiver, which is capable of producing high-precision data largely comparable to those obtained by high-standard geodetic instruments. Thus, it has been proven to be a powerful and cost-effective alternative, not only for monitoring purposes, but also for crustal deformation studies [7]. However the u-blox M8T has an even more convenient price and, in this paper, we demonstrate that it is capable of providing good performance for different applications. In addition, there are several open software programs available today for processing GNSS data and/or real-time GNSS data streams.

The National Institute of Oceanography and Applied Geophysics–OGS (Udine, Italy), in collaboration with the private company SoluTOP SAS (Pavia di Udine, Italy), has developed a cost-effective open GNSS platform called LZER0 based on a u-blox single-frequency M8T receiver (capable of tracking GPS/QZSS, GLONASS, BeiDou, and Galileo constellations) connected to a Raspberry Pi Single-Board Computer (SBC). LZER0, introduced in [8], is a single-frequency GNSS device originally developed for real-time cadastral applications. However, owing to its highly flexible design, LZER0 can also be used in other fields, such as real-time monitoring systems or automotive applications.

In this paper, we first introduce the “base” LZER0 platform and provide a detailed explanation of the hardware and software. We then present three different applications—cadastral, monitoring, and automotive—and describe, for each of them, the corresponding modifications that have been made to LZER0, mainly in terms of the enclosure, the power supply, and the communication system. Finally, we discuss the advantages and limitations, as well as the ongoing developments, of LZER0 and make suggestions for further uses. 

## 2. The LZER0 Platform 

The LZER0 platform is based on four main components: (i) a cost-effective GNSS receiver board with an antenna; (ii) a Raspberry Single-Board Computer (SBC) with Linux on board (usually Raspberry Pi Zero); (iii) RTKLIB [9] software for GNSS elaborations; and iv) shell scripts to manage the GNSS device and RTKLIB. All the material concerning LZER0 (code, technical sheets, and schematics) is available on GitHub (https://github.com/zuliani71/LZER0, accessed on 8 July 2022). 

In general, a stand-alone GNSS receiver can achieve precision of a few meters, but several strategies can be considered to improve its precision (cm or, better, mm, [10,11]). The main strategies used are: Absolute positioning:Post-Processed Precise Point Positioning (PP-PPP, [12,13]);Real-Time Precise Point Positioning (RT-PPP, [14,15]);Relative positioning with the Double Differences (DD) technique:Post Processed Kinematic (PPK, [15,16]);Real-Time Kinematic (RTK [17,18]).

We recall here that absolute-positioning PPP techniques require additional information about precise satellite orbits and clocks from external agencies. Relative positioning DD methods, instead, require both stand-alone GNSS receiver (identified as the Rover) data and GNSS data from another GNSS reference receiver (called the Master, [17]), which increase the cost of the measurement system. However, considering that the RTK algorithms converge faster than those from RT-PPP [19,20], even though RTKLIB can implement both strategies, we chose to use PPK and RTK (DD) techniques, which are more compatible with our GNSS data processing [21,22] and the RTK services provided by the GAMIT/GLOBK [23] and GNSMART [24] software.

We used a very common and cost-effective SBC called Raspberry Pi, especially the “Zero” version, which is small and uses little power, but is powerful enough to support the use of a Linux Operating System (O.S.), and has all the connectivity needed to access a network, such as Wireless Fidelity (Wi-Fi) and BlueTooth (BT). The software chosen to handle the GNSS chipset was demo5 rtklibexplorer, forked from the original RTKLIB (both available on GitHub). Demo5 is a version based on the original RTKLIB 2.4.3, but optimized for low-cost single and dual-frequency GPS receivers (mainly u-blox receivers). We wrote our own code using Linux shell scripts to manage the RTKLIB, and we developed a web interface based on node.js, JavaScript, HTML, and PHP code running on a remote server that can display the real-time results generated by LZER0. For monitoring purposes, we developed shell scripts to generate automatic reports sent via email. Maps and reports were generated using the Generic Mapping Tool (GMT) and the Quantum Geographic Information System (QGIS).

### 2.1. Structure

The LZER0 platform is an evolution of the earlier LONE device, which was developed with a minimal number of components and described in [25]. While LONE was intended to be a simple experiment, LZER0 is intended to be the final product brought to market.

The name LZER0 is composed of the letter “L”, denoting the carrier frequencies in the best-known system, GPS (e.g., L1 and L2), and the word ZER0 to indicate that we used the Raspberry Pi model “Zero”. The reason why the last character of the name is the number “Zero” is twofold:To emphasize that the platform can track different carrier frequencies (e.g., L1, L5 for GPS satellites), regardless of the technology used or the constellations tracked (GPS, GLONASS, Galileo, BeiDou, SBAS, or others);To indicate that it was designed to consume as little power as possible.

The basic principle that guided us in the realization of LZER0 was to keep the price low (at least one-tenth of that of top-of-the-line products), but still provide enough performance and flexibility for the goals we set. In other words, we wanted to develop an instrument that could be used not only for surveying measurements in support of geophysics campaigns, but also for monitoring projects (e.g., landslide monitoring or structural monitoring of damaged bridges or dams) that could be used for both real-time applications (coseismic displacements in seismology or guidance systems for the automotive market) and long-term analysis (crustal deformation studies in support of seismology). We chose a cost-effective instrument to extend the coverage of the measurement points by a factor of ten compared with high-level systems, reducing the need for interpolation for detailed spatial analysis. In this way, it is possible to use real measured values instead of values estimated by mathematical methods at a very reasonable price. 

The LZER0 devices are equipped with an SBC that can run a Linux O.S. We chose the Raspberry Pi family for two reasons: It is well-supported both in terms of hardware (several companies make expansion boards, also called Raspberry Pi hats, for any kind of application) and software (a large community develops applications and publishes them on well-known development platforms, such as GitHub);The cost of buying a Raspberry unit is low (from 5 EUR for the Raspberry Pi Zero base board without Wi-Fi and BT). The chosen model is a Raspberry Pi Zero W, which integrates Wi-Fi and BT hardware, keeping power consumption very low, but providing enough access channels to the hardware.

The LZER0 device was designed to be highly expandable with other sensors (e.g., accelerometers or weather sensors) and expansion boards, but also with other processing methods to integrate multiple-signal knowledge about the measurement point. We chose a Linux-supported SBC to be able to perform complex information manipulation and processing directly on the device itself without the need for a remote server. In this way, we achieve two different goals. The first was to obtain a complete, independent, and decentralized hardware node that could be deployed within a monitoring network, be able to process GNSS signals instantly, and provide a direct measurement of the monitored phenomenon in real time. This is particularly effective for early warning purposes. The second goal was to free the server from multiple processing tasks to improve the overall scalability of the monitoring system and easily add complexity to the topology. This approach, where the decentralized nodes are also used for computation, is innovative compared with previous approaches, where computation is always performed by a single centralized server (this concept can also be seen in other disciplines, such as [26]). In our new vision, the data and products are prepared on remote nodes and the server is used only for their distribution. Therefore, any server failure does not affect the whole process, as the remote nodes would continue to process the results and save them in their internal storage archives, making the monitoring system more robust and reliable.

So far, we have customized the LZER0 device into three different models:Cadastral LZER0, for cadastral applications;Automotive LZER0, for georeferenced guidance applications;Monitoring LZER0, for monitoring systems.

Although all are equipped with the same basic hardware and software, the components are housed in different cases and have different shapes for each model to better perform their tasks. 

### 2.2. Hardware

The hardware components present in each LZER0 model are given in Table 1:SBC: Raspberry Pi Zero W is a very suitable Linux board with two USB ports (one for extending USB capabilities and the other for powering the board), Wi-Fi (802.11 b/g/n wireless) and Bluetooth (4.1) modules, and a mini HDMI port for connecting an external display. The CPU is a 1 GHz 32-bit single-core ARM1176JZF-S and the SDRAM memory has a capacity of 512 MB. The board also includes a Broadcom BCM2835 System on Chip (SoC) and a MicroSD slot for storing the O.S. and data. Most I/O interface signals are provided via a General-Purpose I/O (GPIO) connector. We opted for an 8GB MicroSD card, which is a good compromise for storing a full Debian distribution while being small enough to be quickly cloned for all the units needed. The power consumption is extremely low, 160 mA at 5 V, which is equivalent to 0.8 W. The card format is 65 mm × 30 mm (and 5 mm thick);LAN/USB: with the exception of the Cadastral LZER0, which directly uses the USB ports of the Raspberry, the other models are equipped with a combination of a USB hat (Waveshare USB HUB hat (B) with 4x extended USB 2.0 ports) and a USB–ethernet adapter (tp-link model UE200) or with a single hat including both functions (Waveshare ETH/USB hub hat) to extend the connectivity of the LZER0;Power supply and battery: Cadastral LZER0 is designed with an internal circuit for the power supply and a charging system for the internal 2000 mA LiPo battery. All other models include a system (called ATXraspy, see Table 1) for the safe shutdown of the Raspberry Pi Linux O.S. and a voltage controller for battery and solar panel management (model Western WR30 or model Morningstar ProStar-30);LZER0 main board: Cadastral LZER0 integrates the u-blox M8T receiver into a board designed by OGS, which includes all of the other functions described above (power supply, battery charging system, and input/output communication with Raspberry Pi Zero through an UART port). Automotive and Monitoring LZER0s use a USB card developed by OGS in the form of a Raspberry Pi hat that contains the u-blox M8T GNSS receiver. The USB card can be connected to the Raspberry Pi, but also directly to a PC as a convenient portable GNSS receiver;GNSS antenna: considering that it is an essential part of the equipment and must represent a compromise between performance and cost (no calibration parameters provided, see [27]), we selected the following models: TW4721, a single-band GNSS antenna installed in the Cadastral LZER0, and TW3742, a pre-filtered single-band GNSS antenna for the Automotive and Monitoring LZER0 models. More details are provided in Appendix A;Case: we designed the case for Cadastral LZER0 and printed it with a 3D printer, while for Monitoring and Automotive LZER0, we purchased a standard Gewiss GW44427case.

### 2.3. Software

We wrote the scripts for handling the RTKLIB package installed internally on LZER0, as well as the scripts used to display the real-time position time series of the various remote nodes on a local server and make them available to the end-user via a public web server.

For the sake of clarity, the software system is explained from two points of view: the node’s side (Internal Software) and the user’s side (External Software).

#### 2.3.1. Internal Software (Node Side)

The software node runs on the Raspberry Pi SBC, and it consists of:RTKLIB software; although the entire RTKLIB package has been compiled and is fully available on the Raspberry, we only used the following Command User Interface (CUI) Application Programs (AP):str2str: splits input data from a stream into a multiple-stream output;rtkrcv: executes navigation processing in real-time using raw observation data of GPS/GNSS receivers as inputs;rnx2rtkp: reads RINEX OBS/NAV/GNAV/HNAV/CLK, SP3, and SBAS files and computes GNSS receiver positions and output position solutions.Teqc software ([28]; developed by UNAVCO, no longer supported but still functional) to translate the raw data coming from the GNSS board into the RINEX GNSS data format (only GPS and GLONASS data). In the future developments of the platform, we plan to replace Teqc with Anubis (https://gnutsoftware.com/software/anubis/, accessed on 12 October 2022) or GFZRNX (https://dataservices.gfz-potsdam.de/panmetaworks/showshort.php?id=escidoc:1577894, accessed on 13 October 2022) software for RINEX editing, and to use RTKLIB convbin software for the raw to RINEX (GPS, GLONASS, Galileo, and BeiDou data) conversion;tcsh scripts that we wrote to process the data streams coming from the GNSS board connected to the Raspberry Pi via the serial or USB interface, and to run RTK or post-processing engines provided by RTKLIB.

A backup system and a file-compression system are also provided for data input and output.

The data flow of the entire system is shown in Figure 1, while Table 2 provides an overview of the functionality of each script (see Appendix A for more details).

As shown in Figure 1, the LZER0 device can record data not only from the internal GNSS chipset, but also from the external GNSS reference station (the Master). The data can later be used together with the data coming from the LZER0 device itself in the post-processing mode directly on the LZER0 device or on a remote server from which the data can be downloaded and also further processed with software other than RTKLIB (e.g., GAMIT/GLOBK, [23]). 

The outputs of the LZER0 system are provided in two ways: (1) real-time data streams in Radio Technical Commission for Maritime Services (RTCM) format, standard National Marine Electronics Association (NMEA) format, and LLH RTKLIB format (simple text, also called POS format), which contain the station coordinates in latitude and, longitude, and ellipsoidal height; (2) post-processed results on an hourly basis in raw and POS formats.

#### 2.3.2. External Software (Server Side/User Side)

Although the processing of the GNSS data providing the positions of the acquisition stations is conducted on the stations themselves, we believe that a tool to display these data on the server side can be extremely useful. For this reason, we developed a web platform using node.js, JavaScript, PHP, HTML, and QGIS software. 

Figure 2 shows a diagram of the data flow from each remote GNSS site to the final web page provided to end users. Table 3 lists the main scripts used for the real-time web page to monitor a network with a GNSS reference station and one or more GNSS monitoring stations, which are described in detail in Appendix A.

The code can display, on a browser, a real-time graph containing the plane coordinates, ellipsoidal height, GNSS satellite tracking, position solution quality, and type FIX of a GNSS site capable of providing a POS stream over a TCP socket, and it was designed to allow easy access to the real-time products provided by the nodes through an Internet browser. The remote server has no data processing tasks, and all processing takes place on the LZER0 nodes. This solution makes the whole system highly scalable and avoids increasing the load on the remote server when the number of nodes increases.

## 3. Applications of the LZER0 Platform

The LZER0 platform has been extensively tested from the beginning of its development, and the most interesting outcome is the time of the first FIX for the LZER0 platform (monitoring application), which is about 25 s. We involved students in these operations [25]. This allowed the usability of the instrument and software interface to be tested by non-expert users in order to highlight the weaknesses and shortcomings of the instruments. The low cost, ease of use, speed, and flexibility are the main features that make it an excellent teaching tool. 

To date, the LZER0 device has been successfully used in various projects in both the research and private sectors:Two LZER0 devices were used in the project CLARA “Cloud plAtform and smart underground imaging for natural Risk Assessment”, funded by the Italian Ministry of Education, University and Research (MIUR) [29];Six Cadastral LZER0 devices were produced and sold to the Forestry Administration of the cities of Pordenone, Maniago, and Udine in the Friuli Venezia Giulia Region (FVG) in NE -Italy;Three Automotive LZER0 devices were manufactured by our partner (SoluTOP SAS) and sold to the company Terranova srl, which uses them to drive agricultural machinery;One LZER0 device was used by OGS for real-time monitoring tests for the Cazzaso Landslide [11,30];Three LZER0 devices were installed by OGS for landslide monitoring in the Brugnera area (FVG region, [30]). The monitoring service is operated by OGS on behalf of the Regional Civil Protection;Four LZER0 devices were installed by SoluTOP for environmental monitoring carried out for the PromoTurismoFVG company (NE-Italy).

### 3.1. Cadastral Application

We developed the Cadastral LZER0 model as a small autonomous device to be placed on a topographic survey pole (Figure 3). The pole is equipped with a cradle containing an Android tablet with an Internet connection. The tablet runs surveying software (e.g., SmartRTK or Lefebure NTRIP client and Input from merginmaps company) capable of recording the measured points on an internal DB and displaying them on the maps through the graphical interface of the software itself.

The absolute accuracy achieved by the device with the RTK features reaches some centimeters. For the Cadastral LZER0, some results are available in [25] where we compared the positions estimated by LZER0 with the well-known benchmark coordinates provided by the Italian Military Geographic Institute (IGMI), which is in charge of maintaining the national benchmark coordinate network. The results showed that the coordinate differences ranged from 0.007–0.097 m (North component), 0.033–0.051 m (East component), and 0.059–0.165 m (Up component). These values are of the same order of magnitude as those obtained from more expensive geodetic class instruments when geodetic antennas are used, at least for the horizontal components [18,31]; we are aware that the difference in the vertical component can be caused by the non-calibrated antennas adopted for LZER0. To achieve centimeter-level accuracy, LZER0 combines GNSS observations tracked by the u-blox chipset with differential corrections provided by globally available NTRIP services and made available through the Android device. The results, available in standard NMEA and POS formats, are provided via a TCP port accessible through a Wi-Fi connection.

### 3.2. Automotive Application

We designed the Automotive LZER0 model to be suitable for georeferenced guidance systems, and, so far, it has been used to drive agricultural machinery (Figure 4).

We built the electronics to be powered by 24 V, and we reduced the size of the device to install it in the cockpit of the vehicle. As with the Cadastral LZER0, the positions provided by the device are available with centimeter-level accuracy in real-time (at a typical rate of 1 Hz or up to 20 Hz if required), and are captured by the guidance software in standard NMEA format, which the user can view in any suitable application for Android, Apple, Microsoft, or Linux systems. 

### 3.3. Monitoring Application

We designed the Monitoring LZER0 model to be suitable for permanent monitoring, i.e., to continuously track and record ground motions.

As can be seen in Figure 5, compared with the Cadastral LZER0 model, all of the electronics are in a larger enclosure that provides space for a larger power system (e.g., power controllers and larger batteries that work in conjunction with solar panels). We designed a smaller USB electronics board with the Neo M8T and used a minimal number of components for the power supply and USB connection. We placed the remaining components in a standard electronics box (e.g., GW44427, see Table 1). We chose a Western WR30 (or the equivalent PROSTAR30 produced by the Morningstar company) controller for power supply management and a Teltonika external router for Internet communication (e.g., the RUT230 or RUT240 models). 

Three Monitoring LZER0 devices were used for the landslide monitoring system of the village of Brugnera (Figure 6). Brugnera is located in north-eastern Italy and is crossed by the Livenza River, which runs beside one of the village’s main roads. The river caused a landslide near a bend above the road that severely damaged a parking lot and some nearby houses. The local administration intervened by removing the parking lot and a house, while the movement monitoring was transferred to OGS in collaboration with the Regional Civil Protection. The monitoring system consists of one Monitoring LZER0 device, which serves as a reference station and is installed on the buildings of the municipality outside the landslide (BRU1, Master station), and two Monitoring LZER0 devices installed on the landslide body (BRU2 and BRU3, Rovers stations). We remark here that this monitoring system was configured by choosing the same equipment (GNSS receiver and antenna) for all three stations (BRU1, BRU2, and BRU3), and we also highlight that the baselines between the Master and each Rover are short (about 1 km, see Figure 6a). This allows the removal of common delays using the double DD differences (i.e., antenna phase delay and ionosphere delay).

The LZER0 monitoring system provides both real-time data (one measurement per second) and post-processed data (one measurement per hour) regarding displacements at the two landslide points (POS format). 

At the same site, we also installed a similar system from the private company YETITMOVES (previously described in [11]) for cross-checking the data and results. The YETITMOVES system can post-process displacements with a delay of one hour and consumes only 0.3 W, far less than the 1.7 W of LZER0. However, LZER0 is also capable of providing real-time data every second, has more processing power, and is more versatile, providing all the features of a Linux O.S. that are not available on the YETITMOVES node. A preliminary repeatability comparison between LZER0 and YETITMOVES equipment is available in the Appendix A.

During the installation phases, we also performed some experiments with another Master station (UDI2) installed at the headquarters of CRS in Udine, about 58 km (Figure 6b) from Brugnera. The aim was to test the capacity of the system to fix the solution also for a long baseline (UDI2-BRU1). As BRU1 was the reference station for BRU2 (or BRU3), we decided to also use BRU1 as a Rover with the Master reference station in Udine (UDI2). Thus, station BRU1 served both as a Master for stations BRU2 and BRU3 and as a Rover for station UDI2. The two roles performed by BRU1 are completely independent and do not affect each other, so the BRU2 and BRU3 stations can be used to support landslide monitoring, but experiments can also be performed with UDI2 as a Master. In order to fix the ambiguities [32,33] and achieve fast convergence of the fix algorithm, we decided to use both the GPS and GLONASS constellations. We recall here that using both constellations to correct ambiguities exposes the computational algorithm to some known problems caused by GLONASS inter-channel bias [34,35]. One strategy to overcome this problem is to use receivers of the same brand and model for both the Master and Rover roles (https://rtklibexplorer.wordpress.com/2016/04/30/glonass-ambiguity-resolution-identical-receivers/, accessed on 26 October 2022). For this reason, we also equipped the UDI2 station with a twin to that of BRU1 so that the chipsets of the Master (UDI2) and Rover (BRU1) stations were same (u-blox M8T). In addition, in this experiment, we set the “broadcast ionospheric model” option into the RTKLIB real-time engine (rtkrcv) to take into account the ionosphere’s contribution. 

We analyzed a dataset of the calculated positions of BRU1 (with corrections from UDI2) every second from 00:00 of 1 January 2020 to 00:10 of 5 July. Table 4 shows the results of these experiments for the years 2020, 2021, and 2022, with the real-time positions compared with the reference coordinates obtained in a post-processing calculation using data from the first hour of acquisition. Although the standard deviation for some solutions was particularly high (e.g., 0.249 m for the N coordinate in 2022), showing that there were some outliers probably caused by the large distance between BRU1 and UDI2, the average values were only a few centimeters (0.024 m at most for the E coordinate in 2022, the cause of which is probably also related to the reference coordinates). Considering that the recommended baseline for a GNSS monitoring system is about 5 km [25,36,37], these performance test results are encouraging, taking into account the long baseline tested (58 km) and the fact that the solution was obtained with a single-frequency GNSS.

## 4. Final Remarks and Considerations

A new, cost-effective GNSS platform, called LZER0, was designed and implemented by OGS in collaboration with a private company (SoluTOP SAS) with the goal of providing a low-cost, low-consumption GNSS platform for scientific and industrial research applications. The LZER0 instrument was designed as a flexible, open, and complete system that includes the GNSS equipment and the web portal through which both real-time and post-processed results are made available to the end user. Originally designed for cadastral applications only, LZER0 evolved to become part of a complete real-time landslide monitoring system. Moreover, the various projects in which the LZER0 platform has been used so far show that it is flexible, easy to use in terms of both hardware and software, and suitable for use in research or education, or even in the professional sector, as it can be adapted or upgraded to new GNSS chipsets (even dual- or triple-frequency devices) with little effort.

The platform is based on a widely used SBC, the Raspberry Pi Zero, on which a Linux operating system is installed. The GNSS software installed on it is a version of the well-known open-source RTKLIB software, managed and automated with codes (mainly shell scripts) that we developed for this purpose. The SBC is connected via a USB or serial interface to the actual GNSS receiver, which, in our solution, contains the u-blox M8T single-frequency GNSS chipset. The hardware of the GNSS receiver was manufactured into different versions depending on the needs (surveying, monitoring, or automotive), integrated into a motherboard that also contains the power supply, or as a simple USB card. This modular solution, based on open-source software, makes the whole platform extremely flexible and advantageous both for future developments and from a build cost perspective. 

Two new enhancements will soon be available for LZER0. The first is the installation and testing of the BKG Ntrip Client, a multi-stream client program designed for real-time GNSS applications and capable of providing real-time Precise Point Positioning solutions from both RTCM streams and RINEX files. This will then be compared with the solution produced by RTKLIB on the same LZER0. The second improvement concerns the development and distribution of scripts and code that make LZER0 work. The plan is to use a combination of GitHub services, a container solution, and the Ansible tool to update and customize the management software on all active units of LZER0. This will allow the scientific community to further develop the platform and use these systems to densify the areas to be monitored for both landslides and deformations of the Earth’s crust, while also leaving enough room for the private sector to implement and develop new industrial solutions.

The results presented in this paper are the base for further development of LZER0, which will consider a dual-frequency multi-constellation chipset, allowing the analysis of crustal deformation in the framework of seismological studies (e.g., real-time coseismic monitoring and long-term, low-frequency displacements and velocity field measurements).

## Figures and Tables

**Figure 1 sensors-22-08314-f001:**
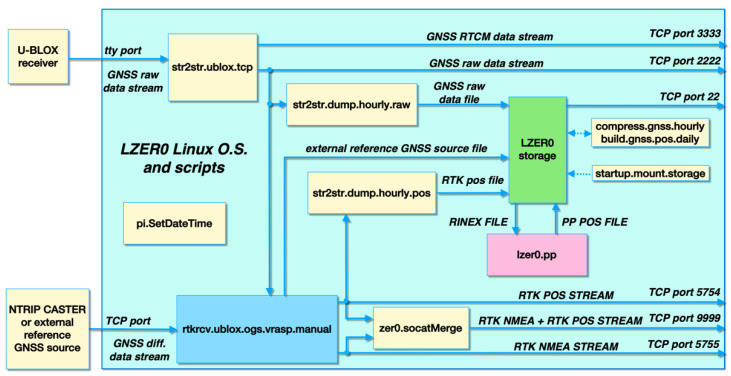
Data flow in the LZER0 instrument (cyan block) and an indication of working scripts (yellow boxes, see also Table 2 and Appendix A). The code captures the raw data coming from the GNSS chipset (in this case, a u-blox model), combines it with GNSS corrections coming from an external source (Networked Transport of RTCM via Internet Protocol–NTRIP–Caster), and produces accurate real-time coordinates. The results are finally made available via various TCP ports.

**Figure 2 sensors-22-08314-f002:**
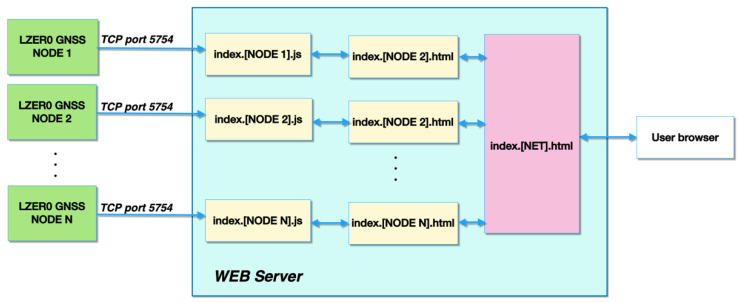
Scheme describing the scripts and code running on the remote server that are used to display the real-time coordinates generated by various remote LZER0 nodes (e.g., the devices of a monitoring network). More details about the scripts and their functions can be found in Appendix A.

**Figure 3 sensors-22-08314-f003:**
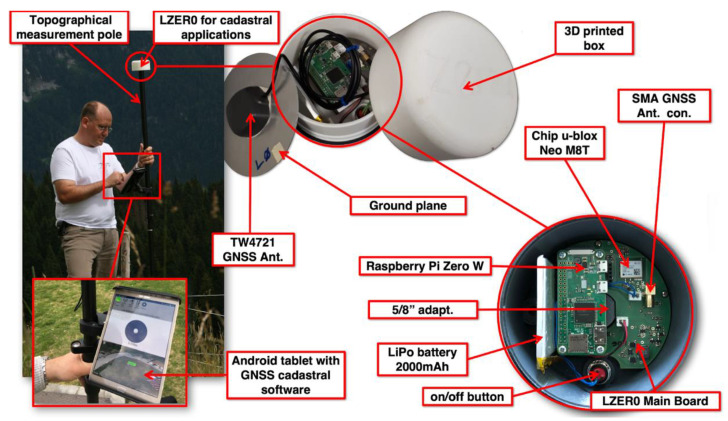
Cadastral LZER0 is a small device designed for surveying or cadastral applications, consisting of the LZER0 motherboard (with the Neo M8T single-frequency chipset), a LiPo battery to power the motherboard (the motherboard can charge the battery via an external USB power supply), and the TW4721 single-frequency GNSS antenna (with an aluminum ground plane). All electronics are included in the 3D-printed case. LZER0 communicates via Wi-Fi connection with an external handheld device (an Android tablet), which is used to track and record the points and trajectories generated by LZER0 in real-time.

**Figure 4 sensors-22-08314-f004:**
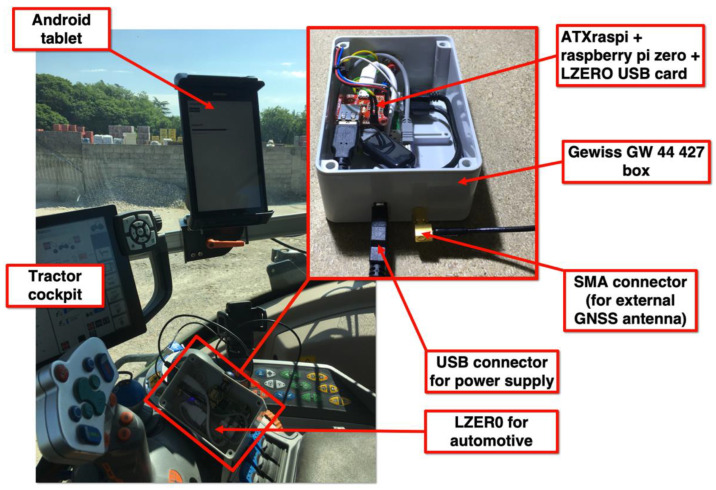
Automotive LZER0 model mounted on agricultural machinery. We installed the electronics in a Gewiss GW44427 industrial box (pictured top center), including the LZER0 USB board, the Raspberry Pi Zero W SBC, and the ATXraspi board. The ATXraspi board controls the power-on and power-off operations for the entire system, including safe shutdown operations for the Linux SBC. The system is powered by a USB plug connected to a 24V–5V converter, and everything is powered by the machinery’s batteries. In the cockpit (on the left side of the picture) of the machinery, above which we installed the GNSS antenna, an Android tablet with Internet access is used to retrieve GNSS corrections from a remote caster and forward them to the LZER0 device. The same tablet is used to display the exact positions of the machinery in real-time and to guide the vehicle.

**Figure 5 sensors-22-08314-f005:**
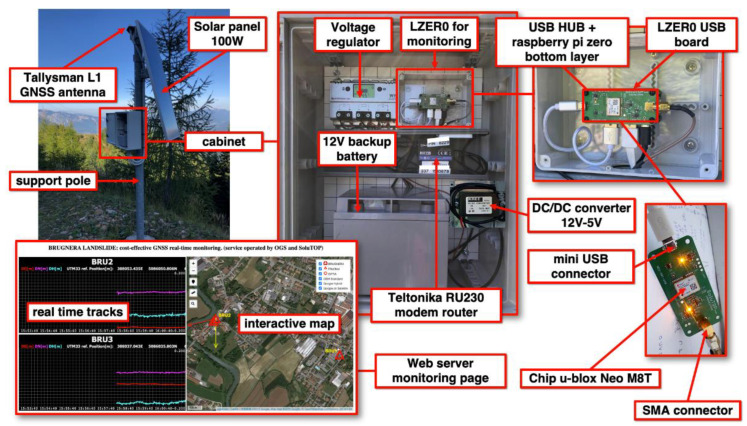
Upper-left corner inset: a typical LZER0 site for a monitoring application. The LZER0 equipment is housed in the cabinet and powered by a solar panel. The LZER0 GNSS antenna is mounted on top of the support mast and is well-anchored to the ground. Top-center picture: contents of the cabinet, including the voltage regulator, backup battery, and LZER0 equipment. Top-right image: the components (mainly the GNSS chipset and the Raspberry Pi Zero SBC) of the LZER0 node. Bottom-right inset: the LZER0 USB board designed by OGS to use the M8T u-blox chipset. On this board, the M8T chip collects the raw GNSS signal from the GNSS antenna connected to the SMA connector and provides the tracked information to the Raspberry Pi SBC USB via the mini-USB connector. Bottom-left: the web page running on the remote server and accessible through an Internet browser. The web page contains, on the left side, the real-time traces of displacements in north–south (NS), east–west (EW), and vertical (UP) directions of each LZER0 node. A map of the LZER0 nodes is shown on the right side of the web page. For each LZER0 node, the horizontal velocities, composed of the NS and EW components (red arrows), and the vertical velocities (yellow arrows), based on the rate of change in the displacement time series, are available. The map is interactive, allowing users to zoom, overlay different background terrain maps, and measure distances and areas.

**Figure 6 sensors-22-08314-f006:**
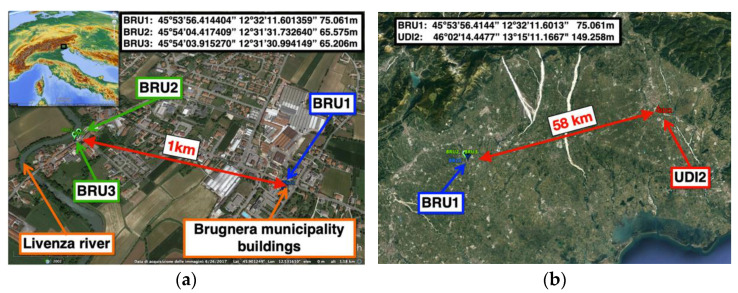
(**a**) Brugnera monitoring system configuration. BRU2 and BRU3 are the monitoring stations (Rovers) and BRU1 is the reference station (Master). (**b**) Performance test with BRU1 as a Rover and UDI2 as a Master station 58 km away.

**Table 1 sensors-22-08314-t001:** Hardware used to manufacture the various LZER0 models, all of them including a Raspberry Pi Zero W with Wi-Fi and BT embedded in the SBC. The motherboard for cadastral applications was developed by OGS and includes the power supply system, u-blox M8T GNSS receiver, power supply, and battery-charging system. For automotive and monitoring applications, OGS developed a USB card in the form of a Raspberry Pi hat that includes the u-blox M8T GNSS receiver. The USB card can also be used directly on a Personal Computer (PC) as a convenient portable GNSS receiver.

Hardware TYPE	Cadastral LZER0	Automotive LZER0	Monitoring LZER0
LAN	No	No	Raspberry hat/USB 2 ethernet
USB	Raspberry Pi ports only	Raspberry Pi ports only	Raspberry hat/USB HUB hat
Power supply	Embedded	External with ATXraspi ^1^	External with DC/DC 12V/5V ^2^
Battery	2000 mAh LiPo battery	External	External
GNSS receiver	Embedded u-blox M8T	U-blox M8T USB card	U-blox M8T USB card
GNSS antenna	Tallysman TW4721	Tallysman TW3742	Tallysman TW3742
Box	3D Printed	Gewiss GW44427	Gewiss GW44427

^1^ https://lowpowerlab.com/guide/atxraspi/, accessed on 26 October 2022. ^2^ KREE converter.

**Table 2 sensors-22-08314-t002:** Scripts that control the internal software system of the LZER0 device. For more details, see Appendix A.

Script Name	Input	Output	Use/Functionality
pi.SetDateTime	NTP server source	O.S. date and time	Setting up O.S. date and time
startup.mount.storage	Non-mounted dev	Mounted dev	Safe mounting of USB storage
str2str.ublox.tcp	U-blox raw data (serial port)	U-blox raw data on TCP/IP ports (2222 and 3333)	Splitting u-blox serial port to TCP/IP streams
rtkrcv.ublox.ogs.vrasp.manual	TCP/IP ports (2222 or 3333) CASTER streams	Precise coordinates on TCP/IP ports (5754 and 5755) and hourly POS file	Real-time coordinate calculus
lzer0.socatMerge	TCP/IP ports (5754 and 5755)	Precise coordinates on TCP/IP port 9999	Merging 5754 and 5755 TCP/IP streams
str2str.dump.hourly.raw	U-blox raw data stream on TCP/IP port 2222	Raw data file	Converting real-time data into file
compress.gnss.hourly	Raw data file	Compressed raw data file	Data file compression
build.gnss.pos.daily	Hourly POS data file	Daily POS data file	Creation of daily POS file
lzer0.pp	Raw data file	Hourly POS data file	Post-processing Coordinate calculus

**Table 3 sensors-22-08314-t003:** Scripts that control the external software on the server side ([NODE #] is the reference identifier of a node). See Appendix A for more details.

Script Name	Input	Output	Use/Functionality
lzer0.forEverStart	List of node.js scripts	Node.js daemons running for each GNSS remote site	Starting node.js daemons for each GNSS remote site
index.[NODE #].js	http requests from users	GNSS site real-time coordinates on the http protocol	Capturing real-time coordinates from remote GNSS sites
index.[NODE #].html	GNSS remote site real-time coordinates on http protocol	web page chart in real-time Coordinates from GNSS remote site	Displaying GNSS remote site real-time coordinates and information
index.[NET].html	list of index.[NODE #].html for each network [NET]	Final web page with real-time charts and GNSS site map	Enabling the final web page for each network [NET]

**Table 4 sensors-22-08314-t004:** The first column contains the years of the available records (hourly POS files with a sampling rate of 1 s). The second column shows the FIX percent (the availability of a FIX solution is an indicator that the determined position is reliable) of reliable solutions estimated for each year. The low fix percentage was due to (1) the experimental regime of the UDI2 station with some long data interruptions, resulting in the absence of the differential data from the UDI2 station; (2) the percentage of solutions with unresolved ambiguities (FLOAT condition); and (3) the fact that all positions with fixed ambiguities but with less than 10% available solutions are discarded within an hour, as this condition is synonymous with poor reliability. Columns 3, 5, and 7 show, respectively, the differences between the mean values of the north–south, east–west, and vertical (E, N, and H) coordinates (EPSG: 32632–WGS 84/UTM zone 32N) estimated for each annual dataset and the reference coordinates obtained in a post-processing calculation using data from the first hour of acquisition. Columns 4, 6, and 8 show the relative standard deviation (DE, DN, and DH). All the reported quantities are in meters.

YEAR	FIX RATIO (%)	E	SDE	N	SDN	H	SDH
2020	53.7	0.013	0.108	−0.003	0.094	0.002	0.185
2021	19.9	0.012	0.086	−0.002	0.103	0.003	0.183
2022	32.5	0.024	0.229	−0.009	0.249	−0.006	0.374

## Data Availability

The documents and materials (schematics, code) used during the development of LZER0 are available on GitHub at https://github.com/zuliani71/LZER0/, accessed on 26 October 2022.

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
