# Peer review of "LZER0: A Cost-Effective Multi-Purpose GNSS Platform"

_sensors, 2022, doi:10.3390/s22218314_

Round 1

Reviewer 1 Report

Plase see attached document

Author Response

The Reviewer's comments are written in blue, whereas the answers are written in black.

Please, notice that when we refer to the Lines in the revised version of the manuscript, we refer to the sensors-1970880.rev.docx version of the manuscript and to the Suppl_Material.rev.docx of the Supplementary Material.

I have carefully read the communication manuscript entitled "LZER0: a cost-effective multi- purpose GNSS platform", which presents a new system for cadastral surveys, ground deformation, structure monitoring and automotive control. Systems of this type exist in great variety. The authors' proposal is the design of a system at very low cost.

Broad comments:

Although the work is well done, the manuscript suffers from some failures. The work, generally speaking, is well designed and conducted. The introduction is adequate to fit the state of the art up to date. The referenced bibliography is appropriate.

The methodology is explained and checked. The weakest part of the article is the low precision obtained if we compare it with other existing methods.

In order to accept this manuscript for publication, some mayor questions could have been made to clarify the explanations and to present the results obtained. In this sense, some aspects are presented on a document for the authors that, in the opinion of this reviewer, should be considered.

We thank Reviewer #1 for the comments and suggestions that helped us in improving the manuscript.

In the following lines, we provide answers to the Reviewer's comments: and in particular we took care of the comment related to the precision issue:

1.- Please check the maximum Keywords accepted.

Thank you, we checked and we selected these ten keywords in the revised version of the manuscript:

Lines: 22-23: “Keywords: GNSS; cost-effective; u-blox; monitoring; real-time; RTKLIB; Raspberry Pi; landslide; cadastral; automotive.”

2.- Line #42, please locate the OGS Institute (place and country). Probably Italian reader will know it, but this journal has a worldwide distribution and not every possible reader will know the location of the institute.

We added the information required in the revised version of the manuscript at:

Lines: 48-49: “The National Institute of Oceanography and Applied Geophysics – OGS (Udine, Italy),…”.

3.- Abstract: Line #13. Please change ‘topographic measurements’ for ‘surveying measurements’.

Done.

Line: 13 “…, suitable not only for surveying measurements…”

4.- Line #83. Please indicate how the reader of this manuscript can access this demo5 version of RTKLIB: repository, website, data server, request to authors...

Since in the main text is already specified the software is already publicly available on GitHub, we added the demo5 link in the Acknowledgments:

Lines: 556-557: “…and information on demo5 version is available at https://github.com/rtklibexplorer/RTKLIB/, accessed 18 October 2022;”.

5.- Lines #89-90. Please add appropriate citation to GMT and QGIS. Two new paragraphs on the References section must be completed.

References for these software are reported in the Acknowledgements at Lines 557-559 of the revised version of the manuscript. We added [38] to complete the information at Line 663.

6.- Line #107. Please consider comment #3. This confusion is extended several times throughout the entire manuscript. Please correct.

Done. We correct them throughout the manuscript.

7.- Line #123, ‘5 euro’

Done. We corrected the typo (Line 136 of the revised manuscript).

8.- Table 1: We can eliminate the lines that are repeated for all the tests. 'SBC' and 'WiFi and BT' can be removed from the table and commented out in the main text. This way the table is more readable.

Done. The common information among the different LZER0 models is now specified in the caption of Table 1 in the revised manuscript.

Lines 166-167: “Hardware used to manufacture the various LZER0 models, all of them including a Raspberry Pi Zero W with Wi-Fi and BT embedded in the SBC.”

9.- Section 2.2 Hardware: the authors describe six hardware elements. Although later in the text, some general images are included where they more or less appear, in this section it is necessary to include an image of each of these elements in which each of the elements appears by itself, without mixing it with other elements or cables. Please include.

We added the images in the Supplementary Material section C.

10.- Line #189: we need to know something more about the antennas that are being used. Especially its calibration. If we are going to look for a high accuracy in the tests, this is important. Please add this information.

We agree with the Reviewer that the calibration is important, unfortunately for these low-cost antennas the calibration is not provided (as also indicated in [27]) and we recognize that this is their weak point especially for the measurement of the vertical component. These limitations are however indicated in [25]. A solution, to overcome this limitations, is to use short base-lines (O’Keefe et all. 2016 reported in the Supplementary Material References) and the same antennas on both Rover and Master sites (Hofmann-Wellenhof 2001). This is what we have implemented in the Monitoring LZER0 system. In this way the DD technique would allow a substantial reduction of the phase delay (Hofmann-Wellenhof 2001 reported in the Supplementary Material References). In the manuscript for sake of completeness, we have also added the references to the data sheets of the low cost antennas (Tallysman TW3742, TW4721) used.

At the beginning of our study we firstly selected   very low cost antennas: ARKNAV A-130, GAACZ-A, ANT-1575R. Unfortunately, they are not provided with robust interference rejection systems (usually because of the filter implemented in their Low Noise Amplifier LNA), hence the field tests we carried out showed  intermittent and generally unstable behavior when they are used in combination with the RTKLIB software. Therefore, we considered  more expensive antennas (Tallysman TW4721 and TW3742), that are  adopting  more performing LNA,  capable of rejecting the interference produced by other electromagnetic systems such as the Long Term Evolution (LTE) and the WiFi. The results that we obtained showed   higher stability in the tracking of satellites. We therefore decided to use them  as the best compromise between cost, performance and dimensions (the latter is especially important in the Cadastral LZER0 where the designed box has modest dimensions). We added these information in the Supplementary Material Section C and a citing it in the revised version of the main manuscript, at:

Lines 205-213: “GNSS antenna: considering that it is an essential part of the equipment and must represent a compromise between performance and cost (no calibration parameters provided see at [27]), we have selected these models: TW4721, a single-band GNSS antenna installed in the Cadastral LZER0, and TW3742, a pre-filtered single-band GNSS antenna for the Automotive and Monitoring LZER0 models. More details about the TW3742 and the TW3742 antennas are provided in the Supplementary Material Section C”.

11. Line #214. The RINEX3 version is becoming a standard that will very quickly replace RINEX2. How is this going to be handled with TEQC? Please comment.

We thank the Reviewer for the interesting question. Indeed, we are facing this problem also for our  GNSS permanent site network with high cost sensors. Based on our experience, we think that the best strategy is to convert the data from the u-blox to RINEX v3 format using the RTKLIB applications called rtkconv (CUI version) or convbin (AP version), and then to edit the RINEX header according to the users’ needs using gfzrnx or also Anubis. We modified the text of the revised version manuscript as follows:

Lines: 234-242: “Teqc software ([28]; developed by UNAVCO, no longer supported but still functional) to translate the raw data coming from the GNSS board into the RINEX GNSS data format (only GPS and GLONASS data). For the future developments of the platform we plan to replace Teqc with Anubis (https://gnutsoftware.com/software/anubis/, accessed on 12 October 2022) or GFZRNX (https://dataservices.gfz-potsdam.de/panmetaworks/showshort.php?id=escidoc:1577894, accessed on 13 October 2022) software for the RINEX editing, and to use RTKLIB convbin software for the raw to RINEX (GPS, GLONASS, Galileo and BeiDou data) conversion.”

12.- Line #300. What is ‘merginmaps’ ?

We clarified that merginmaps is the company providing SmartRTK or Lefebure NTRIP client in the Line 332 of the revised version of the manuscript. Additional information was anyway available in the Acknowledgements of the revised manuscript at:

Lines: 562-563 “information on Input from merginmaps can be found at https://merginmaps.com/, accessed on 8 July 2022;”

13.- Line #312. The authors say they achieved and absolute accuracy of a few centimeters. The precision that is finally obtained is perhaps the most determining parameter when evaluating the quality of an equipment or a measurement methodology. Even cadastral works are subject to a minimum level of precision. We need to know exactly what precision the authors have obtained with their equipment in the cadastral application that is evaluated in this section. In addition, we need to know if the precision value that the authors provide comes from a repeatability study or has been obtained by comparison with another more precise standard measurement methodology. Please complete all these parts.

We thank the Reviewer for this comment. We modified the text and we added more information about the accuracy of the Cadastral LZER0 in the main text of the revised manuscript:

Lines: 344-354) “The absolute accuracy achieved by the device with the RTK features is of some centimeters. For the Cadastral LZER0 some results are available in [25] where we compared the positions estimated by LZER0 to the well-known benchmark coordinates provided by the Italian Military Geographic Institute (IGMI) which is in charge to maintain the national benchmark coordinate network. The results show that the coordinate differences range 0.007-0.097 m (North component), 0.033-0.051 m (East component), 0.059-0.165 m (Up component). These values are of the same order of magnitude as those obtained from more expensive geodetic class instruments when geodetic antennas are used, at least for the horizontal components [18, 31], we are aware that the difference in the vertical component can be caused by no-calibrated antennas adopted for LZER0.”

Furthermore we added the preliminary results of the repeatability test (using the standard deviation as a measure of the precision), obtained with Monitoring LZER0, in the revised version of the Supplementary Material Section B (see also answers to question #19). They show that values (taking into account a 2 sigma constraint) in the order of  +/- 8mm for the North component, +/- 6mm for the East component and +/-16mm for the Up component are obtained.

14.- Lines #312-314. The authors say the achieved and absolute accuracy of a few centimeters and that this is comparable to that of more expensive geodetic class instruments when geodetic antennas are used. Although based on a single citation, this is very hard to believe when geodetic

equipment has various frequencies and calibrated antennas that maintain this accuracy to within 1-2 cm. A simple literature search would confirm this. What the authors say is not true and this line has to be removed from the manuscript, due to it is not true.

We thank the Reviewer for this critical comment. As you stated, using calibrated antennas allows accuracies within 1-2 cm and LZER0 has a lower accuracy (we have mentioned [25] to give an idea of ​​its accuracy for the survey version). We consider that "comparable" is probably not the right term to use. We changed the sentence and we added a new reference ([31]).

Line: 350-354 “These values are of the same order of magnitude as those obtained from more expensive geodetic class instruments when geodetic antennas are used, at least for the horizontal components [18, 31], we are aware that the difference in the vertical component can be caused by no-calibrated antennas adopted for LZER0.”.

15.- Line #322. ‘Agricultural tractor’ to ‘Agricultural machinery’. Please change it overall text.

Done.

16.- Table 4. Line #403. It seems that what is in the table are not UTM zone 32 coordinates, but rather coordinate differences. If so, this must be corrected in the table caption and in the table headers.

We thank the Reviewer. We clarified the sentence in the caption of Table 4.

17.- Table 4. If all units represented here are meters, (m) should be removed from the table headers. In the caption, a line of text must be included at the end indicating that all the quantities in the table are meters. Please do.

18.- Table 4. If the columns to the right of the coordinates (or differences) are standard deviations, it is preferable to label the headers 'SD' or 'sigma' rather than 'D'. The reader may confuse ‘D’ with a difference in values instead of an indicator of uncertainty and generate further confusion. Please correct.

Done. We changed Table 4 and its caption according to the Reviewer’s suggestions.

19.- Section 3.3 The monitoring application. The authors comment that another system called YETITMOVES has been installed in the same monitoring sites to crosscheck the results. The authors have forgotten to include the results of this comparison. This is very important because it is an independent evaluation of accuracy with another methodology, which makes it very interesting. Please complete with the results of this comparison.

We added a comparative result regarding the repeatability in the revised Section B of the Supplementary Material document and we included a sentence in the revised version of the manuscript:

Lines: 435-436 “A preliminary  comparison on the results of repeatability tests between the LZER0 and the YETITMOVES equipment is available in the Supplementary Material (Section B).”

Since we agree with the Reviewer that this topic is of absolute importance, we think that it should be examined in depth into a dedicated article, where data and contents can be properly elaborated. The aim of this paper is just to describe the development of a new device which has the possibility of being applied in different areas of application, though we are aware of the fact that, since it is subjected to continuous development, the current version cannot be fully exhaustive in every detail. However, as suggested by the reviewer, we think that a comparison, even preliminary, can surely improve the value of this paper (see also answers to question #13).

20.- Section 3.3 The monitoring application. Standard deviations achieved in table 4 invalidate the results obtained for monitoring. This is the weakest part of the manuscript and here we have a big problem, because the authors have not been able to justify the application of monitoring with the results they have obtained, since they are not conclusive. From my point of view, the authors here have two possibilities: discard this application and keep the manuscript only with the other two applications that the authors do seem to demonstrate. Other possibility, repeat the validation of the monitoring application results using another methodology whose standard deviation o uncertainty is less than the value to be measured, as is usually done in metrological applications. Please select what is finally going to be done with this section.

We thank the Reviewer for this important observation. The monitoring system presented in this section is formed by BRU2 and BRU3 stations coupled with Master station BRU1 located a few km away. The three sites (BRU1, BRU2, BRU3) have been configured by choosing the same systems on all the three stations and short baselines (for the removal of common delays using the double DD differences) in order to overcome system limitations. We show that this real-time monitoring system using LZER0 is capable of fixing the solution and it can be used also for monitoring purposes. Since we agree with the reviewer that including validation tests would makes this section more reliable we added the description of a test we carried out to the manuscript. This  test allows to evaluate the repeatability (see also answers to question #13, #14 and #19) of the results obtained by the proposed  system (i.e.  using Monitoring LZER0 ) compared to that of the commercial system YETITMOVES. The results of the test are shown in the revised version of the Supplementary Material Section B.

The test using UDI2-BRU1 baseline has been carried out to test the capacity of the system of fixing the solution also for long baselines. It is remarkable that the system was able to fix the solution even in these conditions, although we agree with the Reviewer that the quality of the results (see Table 4), in terms of position estimates, is low ( depending, for example, on the type of antenna used and on the long baselines which introduce errors in the evaluation of ionospheric delay, especially in single frequency systems). We modified the text of the revised manuscript to better clarify our point of view at:

Lines: 401-406 “We remark here that this monitoring system have been configured by choosing the same equipment (GNSS receiver and antenna) on all the three stations (BRU1, BRU2 and BRU3) and we also highlight that the baselines  between Master and each Rover is short (about 1 km, see Figure 6 a). This allows the removal of common delays using the double DD differences (i.e. antenna phase delay, ionosphere delay).”;

Lines: 435-436 “A preliminary repeatability comparison between LZER0 and YETITMOVES equipment is available in the Supplementary Material (Section B).”

Lines: 459-460 “The aim is just to test the capacity of the system of fixing the solution also for a long baseline (UDI2-BRU1).”

Reviewer 2 Report

Dear Authors,

Comments on the study are given in the attached PDF file. I also congratulate you on this platform you have developed. The platform offers solutions to many geodetic applications at a low cost.

Author Response

The Reviewer's comments are written in blue, whereas the answers are written in black.

Please, notice that when we refer to the Lines in the revised version of the manuscript, we refer to the sensors-1970880.rev.docx version of the manuscript and to the Suppl_Material.rev.docx of the Supplementary Material.

Comments on the study are given in the attached PDF file. I also congratulate you on this platform you have developed. The platform offers solutions to many geodetic applications at a low cost.

We thank Reviewer #2 for the comments and suggestions that helped us in improving the manuscript.

In the following we provide answers to the Reviewer's comments:

The study is written in a fluent and easy-to-understand manner. But, this study is a communication-type article that introduces a platform that hosts a low-cost GNSS receiver and an internet-connected computer and enables real-time and post-process positioning. The low-cost platform is developed by the National Institute of Oceanography and Applied Geophysics, in collaboration with a private company (SoluTOP SAS) in Italy. Actually, this study is not really a scientific type of article and does not contain any analysis.

We think that the content of the manuscript is appropriate for the Journal “Communications” indeed. This kind of paper includes state-of-the-art methods or experiments as well as the development of new technologies or materials. We finally added a preliminary repeatability comparison between LZER0 and YETITMOVES equipment in the Supplementary Material Section B as indicated in the revised manuscript at Lines: 435-436.

Abstract:

A. In the abstract section, it is necessary to explain what type of positioning is done using RTKLIB software, absolute positioning or relative positioning, post-process or real-time, PPP or PPK.

B. In addition, it should be stated in the abstract section that the low-cost GNSS receiver (M8T) used in the study is single-frequency.

We modified the abstract of the revised manuscript adding the information suggested:

Lines 14-18: “The LZER0 platform is a complete system that includes the GNSS equipment (M8T single frequency model produced by u-blox) and the web portal where results are displayed. The GNSS data are processed using the RTKLIB software package, and the processed results are made available to the end user. The relative positioning mode was adopted both with real-time and post-processing RTKLIB engines.”

Introduction:

C. Although the authors used the u-blox M8T low-cost GNSS receiver in this study, they mention the u-blox F9P receiver in the introduction, which is confusing. Authors should cite which study used the u-blox F9P low-cost GNSS receiver.

The introduction aims to illustrate the state of the art of the cost-effective GNSS instrumentation. In order to clarify the sentence, in the revised version of the manuscript, we added the single-frequency receivers included in the cited references and specified the study using a dual-frequency cost-effective device (F9P):

Lines: 35-46: “Fortunately, in recent decades, significant advances in GNSS technology have enabled the development of low-cost single-frequency sensors (i.e. u-blox EVK-5T, LEA-4T, LEA-6T, NEO-7P, M8T, EVK-6T) that are attracting the scientific community's interest, and several studies have been conducted showing that, equipment built with them, are capable of centimeter-scale positioning [1-6]. A recent study shows also the excellent performances of the cost-effective dual-frequency u-blox F9P receiver, which is capable of producing high-precision data largely comparable to that obtained by high-standard geodetic instruments. Thus, it proves to be a powerful and cost-effective alternative, not only for monitoring purposes but also for crustal deformation studies [7]. However the u-blox M8T has an even more convenient price and, in this paper, we show it is capable of providing good performances for different applications.”

D. In this section, some technical specifications of the low-cost M8T GNSS receiver should be given (for example: Is it single or multiple frequencies, which satellite systems can it use?), and how the connection with Raspberry SBC is made and in which data format (raw, RINEX, RTCM, etc.) they share should be explained.

We thank the Reviewer for the suggestions. We added the required details regarding  M8T in the introduction.

Lines: 50-52: “… based on a u-blox single-frequency M8T receiver (capable of tracking GPS/QZSS, GLONASS, BeiDou, Galileo constellations) and connected…”

We think that the description about the connection with the Raspberry SBC and the data format, that changed in the different versions of LZER0, fits better in the Hardware and Software Sections 2.2 and 2.3 and therefore we decided to keep it there.

E. In the introduction, application examples containing only low-cost GNSS receivers should not be given, but also information about other platforms prepared for special purposes in this way should be given and cited from the literature.

We understand your comment, however, we think that the references that we cited in the Introduction (from 1 to 6) provide a description of both the used GNSS chipset and, also, of the platform that is implementing that chipset. In fact, without that specific platform, the chipset is not usable. We modified the following lines in the revised version of the manuscript:

Lines 35-30: “Fortunately, in recent decades, significant advances in GNSS technology have enabled the development of low-cost single-frequency sensors (i.e. u-blox EVK-5T, LEA-4T, LEA-6T, NEO-7P, M8T, EVK-6T) that are attracting the scientific community's interest, and several studies have been conducted showing that, equipment built with them, are capable of centimeter-scale positioning [1-6].”

The LZER0 platform

F. This GitHub (https://github.com/zuliani71/LZER0/) link is not active.

We apologize for the slip. The link is now active.

G. The main positioning strategies that can be realized with the RTKLIB software should first be listed as absolute and relative positioning. After that, Real Time-Precise Point Positioning (RT-PPP) and Post Processed-Precise Point Positioning (PP-PPP) options can be said for absolute positioning. For relative positioning (positioning using DD), a single baseline Real Time Kinematic (RTK) or Post Processed Kinematic (PPK) technique should be mentioned.

Following the Reviewer’s suggestions we modified the following lines in the revised version of the manuscript:

Lines 73-80: “The main strategies used are:

  • absolute positioning:
    • Post Processed Precise Point Positioning (PP-PPP, [12, 13]);
    • Real-Time Precise Point Positioning (RT-PPP, [14, 15]);
  • relative positioning with Double Differences (DD) technique:
    • Post Processed Kinematic (PPK, [15, 16]);
    • Real-Time Kinematic (RTK [17, 18]).”.

H. Relative positioning is more precise but not more comfortable because it requires at least one Master receiver other than the Rover receiver, which increases the cost. As an alternative to relative positioning, the absolute positioning method, PPP, which has been popular in recent years and is frequently used in practical and scientific studies, develops over time. RT-PPP technique can be performed using precise products served by IGS- RTS with the help of a computer connected to the Internet. This issue should be addressed in the second part of the study (Lines 75-76).

We agree with the Reviewer that relative positioning is more precise and we also described its limitations in the text (e.g. the need of a Master receiver which increases the cost). We avoided the use of PPP because it requires 1) more time to converge (that might not be the optimal option for several real-time applications), 2) an external service for clock and orbit corrections and, as also stated by the Reviewer, is nevertheless less precise of relative positioning. In order to better clarify these issues, we slightly modified the revised version of the manuscript as follows:

Lines: 81-90: “We recall here that absolute positioning PPP techniques require additional information about precise satellite orbits and clocks provided from external agencies. Relative positioning DD methods, instead, require both stand-alone GNSS receiver (identified as the Rover) data and GNSS data from another GNSS reference receiver (called Master, [17]) which increase the cost of the measurement system. However, considering that the RTK algorithms converge faster than those from RT-PPP [19, 20], even though RTKLIB can implement both strategies, we chose to use PPK and RTK (DD) techniques that are more compatible with our GNSS data processing [21, 22] and RTK services provided by GAMIT/GLOBK [23] and GNSMART [24] software.”.

I. It is necessary to close the parenthesis that opens on Line 140. (see this concept also in other disciplines such as [26] ).

Done.

2.2 Hardware

J. Why is there no information about the LZER0 main board in this section? Does the main board provide the relationship between the SBC and the GNSS receiver?

The information was included, but we understand that it was not easy to be retrieved. We clarify the sentence in the revised version manuscript as follows:

Line 198-204: “LZER0 main boardGNSS receiver: Cadastral LZER0 integrates the u-blox M8T receiver into a board designed by OGS, which includes all the other functions described above (power supply, battery charging system, input/output communication with Raspberry Pi Zero through an UART port). Automotive and Monitoring LZER0 use a USB card developed by OGS in the form of a Raspberry Pi hat that contains the u-blox M8T GNSS receiver. The USB card can also be connected to the Raspberry Pi but also directly to a PC as a convenient portable GNSS receiver;”.

2.3.1. Internal Software (node side)

K. The tasks of the three sub-applications of the RTKLIB software given between lines 210-212 should be explained in this section.

We added the required information in the revised version of the manuscript:

Lines 227-231: “

  • str2str: it splits the input data from one stream into a multiple stream output;
  • rtkrcv: it executes the navigation processing in real‐time using the input raw observation data collected by the GPS/GNSS receivers;
  • rnx2rtkp: it reads the RINEX OBS/NAV/GNAV/HNAV/CLK, SP3, SBAS files, computes the GNSS receiver positions and output position solutions.”

and we included the reference to the RTKLIB manual in the Supplementary Material Section A.

L. The platform converts the GNSS raw data from the M8T receiver to rinex v2 format by using teqc software. In this case, only two constellations (GPS and GLONASS) data can be used. For the 3rd and 4th satellite systems, it is necessary to use rinex v3 or v4 format. In this case, can gfzrnx software that supports rinex v3 and v4 formats be used instead of teqc software?

We thank the Reviewer for the interesting question. Indeed, we are facing this problem also for our  GNSS permanent site network with high cost sensors. Based on our experience, we think that the best strategy is to convert the data from the u-blox to RINEX v3 format using the RTKLIB applications called rtkconv (CUI version) or convbin (AP version), and then to edit the RINEX header according to the users’ needs using gfzrnx or also Anubis. We modified the text of the revised version manuscript as follows:

Lines: 234-242: “Teqc software ([28]; developed by UNAVCO, no longer supported but still functional) to translate the raw data coming from the GNSS board into the RINEX GNSS data format (only GPS and GLONASS data). In the future developments of the platform we plan to replace Teqc with Anubis (https://gnutsoftware.com/software/anubis/, accessed on 12 October 2022) or GFZRNX (https://dataservices.gfz-potsdam.de/panmetaworks/showshort.php?id=escidoc:1577894, accessed on 13 October 2022) softwares for the RINEX editing, and to use RTKLIB convbin software for the raw to RINEX (GPS, GLONASS, Galileo and BeiDou data) conversion.”

2.3.2. External Software (server side/user side)

M. There is an excess at the bottom of figure 2 on Line 268.

 Done.

3.3 The monitoring application

N. I wonder if the second of the two BRU1 on Line 427 will be UDI2. "We analyzed a dataset of calculated positions of BRU1 (with corrections from UDI2)"

Yes, correct. Thanks for pointing this out. Following the Reviewer’s comments we modified the sentence:

Lines 478-479: “We analyzed a dataset of calculated positions of BRU1 (with corrections from UDI2)”

O. It is said that the 58 km baseline between UDI2 and BRU1 on Line 436 is resolved with single-frequency GNSS data. How was the ionospheric error removed from the solution in this case? Please explain this.

We set up the rtkrcv engine with the “Broadcast” option for the ionosphere correction, that means applying a broadcast ionospheric model to achieve a fix with a single frequency GNSS receiver. We added this information in the revised manuscript as follows:

Lines 451-453: “In addition, in this experiment, we set the “broadcast ionospheric model” option into the RTKLIB real-time engine (rtkrcv) to correct the ionosphere contribution.”

In the same manuscript we also added the following note:

Lines 459-460: “The aim is just to test the capacity of the system of fixing the solution also for a long baseline (UDI2-BRU1).”

  1. Final remarks and considerations

P. According to Line 461, two new enhancements will be added to the LZER0 system in future studies. As a third enhancement, it is recommended to add the Galileo satellite system to the platform.

We thank the Reviewer for the suggestion. We will consider this in future. We modified a sentence in the text of the revised version manuscript in order to mention this.

Lines 526-530: “The results presented in this paper are the base for a further development of LZER0, which will consider a dual-frequency multi-constellation chip, allowing the analysis of crustal deformation in the framework of seismological studies (e.g., real-time coseismic monitoring and long-term, low frequency displacements and velocity fields measurements).”

Reviewer 3 Report

Dear authors,

This manuscript is well written with good logic flow and structure. It explicitly introduces the hardware and software onboard the LZERO receiver. More importantly, the three main application scenarios are clearly explained. I think the paper can be accepted soon only after a few minor modifications,

1.Is it possible to add some analyses on the tracking performance (Nr.GNSS sats, code/phase residuals, SNR) of this device? It feels like the performances under the three applications might be quite different

2.Fig.2 has a weird upward sign, is it an error from screenshot?

3.Tab.4, how do you know the exact distance/coordinates of the stations?

I am impressed by this work, Thanks!

Author Response

The Reviewer's comments are written in blue, whereas the answers are written in black.

Please, notice that when we refer to the Lines in the revised version of the manuscript, we refer to the sensors-1970880.rev.docx version of the manuscript and to the Suppl_Material.rev.docx of the Supplementary Material.

This manuscript is well written with good logic flow and structure. It explicitly introduces the hardware and software onboard the LZERO receiver. More importantly, the three main application scenarios are clearly explained. I think the paper can be accepted soon only after a few minor modifications.

We thank Reviewer #3 for the suggestions and comments that helped in improving the manuscript. Detailed answers to the Reviewer’s comments are provided as follows:

Is it possible to add some analyses on the tracking performance (Nr. GNSS sats, code/phase residuals, SNR) of this device? It feels like the performances under the three applications might be quite different.

We thank the Reviewer for the suggestion. In this work, although we carried out several tests concerning the difference between known a priori values ​​(for example with the coordinates of official benchmarks calculated by the national topographical institutes), we mainly focused on guaranteeing that the instrument was able to reach the first FIX quickly. Analysis and statistics of the signal-to-noise ratio and other parameters will be the object of further studies. An evaluation of the time of the first FIX for the LZER0 platform (monitoring application), of about 25s, is available in [25]. We modified the text of the revised version manuscript as follows:

Lines 203-305: “The LZER0 platform has been extensively tested from the beginning of its development and the most interesting outcome is the time of the first FIX for the LZER0 platform (monitoring application), which is of about 25s. We involved students in these operations [25].”

We finally added a preliminary repeatability comparison between LZER0 and YETITMOVES equipment in the Supplementary Material Section B as indicated in the revised manuscript at Lines: 435-436.

Fig. 2 has a weird upward sign, is it an error from screenshot?

Thanks for pointing this out. Yes, it was a screenshot problem. We changed figure 2 in the new version of the manuscript.

Tab. 4, how do you know the exact distance/coordinates of the stations?
Tab. 4 reports only the distance between the real-time solution and a more robust post-processed solution produced for the same site. This information can be read from the .pos file produced by the RTKLIB engines (rtkrcv for the real-time solution and rnx2rtkp for the post-processed solution). We clarified this information in the revised version manuscript at:

Lines 479-482: “Table 4 shows the results of these experiments for the years 2020, 2021, and 2022, with the real-time positions compared to the reference coordinates obtained in a post-processing calculation using data from the first hour of acquisition.”

I am impressed by this work, Thanks!

We thank the Reviewer for the very positive statement.

Round 2

Reviewer 1 Report

The authors have made a great effort to improve the original manuscript. I have to thank them for taking into account my comments and suggestions for the initial manuscript. My compliments to them.